# Resistance Evaluation of Dominant Varieties against Southern Rice Black-Streaked Dwarf Virus in Southern China

**DOI:** 10.3390/v13081501

**Published:** 2021-07-30

**Authors:** Shunkang Zhou, Yaling Zhao, Zhenyi Liang, Ruifeng Wu, Biao Chen, Tong Zhang, Xin Yang, Guohui Zhou

**Affiliations:** Guangdong Province Key Laboratory of Microbial Signals and Disease Control, College of Plant Protection, South China Agricultural University, Guangzhou 510642, China; zsk18819455887@163.com (S.Z.); zhaoyalingz@163.com (Y.Z.); 13286532820@163.com (Z.L.); ruifengoffice@163.com (R.W.); chenbiaobcb@163.com (B.C.); zhangtong@scau.edu.cn (T.Z.); yangxin@scau.edu.cn (X.Y.)

**Keywords:** *southern rice black-streaked dwarf virus* (SRBSDV), white-backed planthopper (WBPH), rice virus resistance

## Abstract

*Southern rice black-streaked dwarf virus* (SRBSDV), a *Fijivirus* in the *Reoviridae* family, is transmitted by the white-backed planthopper (*Sogatella furcifera*, WBPH), a long-distance migratory insect, and presents a serious threat to rice production in Asia. It was first discovered in China’s Guangdong Province in 2001 and has been endemic in the south of China and north of Vietnam for two decades, with serious outbreaks in 2009, 2010, and 2017. In this study, we evaluated the resistance of 10 dominant rice varieties from southern China, where the virus overwinters and accumulates as a source of early spring reinfection, against this virus by artificial inoculation. The results showed that in all tested varieties there was no immune resistance, but there were differences in the infection rate, with incidence rates from 21% to 90.7%, and in symptom severity, with plant weight loss from 66.71% to 91.20% and height loss from 34.1% to 65.06%. Additionally, and valuably, the virus titer and the insect vector virus acquisition potency from diseased plants were significantly different among the varieties: an over sixfold difference was determined between resistant and susceptible varieties, and there was a positive correlation between virus accumulation and insect vector virus acquisition. The results can provide a basis for the selection of rice varieties in southern China to reduce the damage of SRBSDV in this area and to minimize the reinfection source and epidemics of the virus in other rice-growing areas.

## 1. Introduction

Rice (*Oryza sativa*) is one of the main food crops around the world, and viral disease is a serious problem in rice production [1,2]. More than 10 kinds of rice viruses have been reported; these are widely distributed in the rice-producing areas of Asia and Africa. Most of the viruses are transmitted by insects and lead to massive losses [3,4].

Since the beginning of the 21st century, the *southern rice black-streaked dwarf virus* (SRBSDV) has occurred in southern China, northern Vietnam, and Japan; it is transmitted by a long-distance migratory pest, the white-backed planthopper (*Sogatella furcifera*, WBPH), and is the most serious rice virus in Asia [5,6,7]. The infected plants are dwarfs, with dark green leaves and curly leaf tips. At the jointing stage, there are aerial fibrous roots and high node branches on the stem nodes of the diseased plants. On the surface of the stem, there are milky white or brownish-black nodular protrusions arranged longitudinally with the appearance of wax dots about 1~2 mm in size. The main yield loss caused by the disease is due to the fact that the diseased plants cannot head when they are infected at the seedling stage [7,8,9]. In recent years, as a result of global warming, WBPH migrates to the north earlier and transmits the virus to infect rice seedlings, which makes the disease more serious [10,11].

Southern China is a major area where SRBSDV overwinters and accumulates in early spring as a source of infection [7]. At present, control of the disease mainly relies on physical barriers and insecticide application to reduce the transmission of WBPHs [12,13,14]. The screening and utilization of resistant varieties is an urgent need in production. However, no resistance evaluation of dominant varieties against SRBSDV in southern China has been reported. In this study, the SRBSDV resistance of 10 dominant rice varieties in southern China was evaluated based on infection rate, symptom severity, virus accumulation, and insect vector virus acquisition by artificial inoculation, which provides a basis for the selection of varieties to reduce the disease damage and minimize the reinfection source.

## 2. Materials and Methods

### 2.1. Rice, Virus, and Vector Insects

A total of 10 dominant rice varieties—hybrid indica rice Guang8you 165, Baixiangyou 9978, Guyou 248, Yexiangyou 9, and Guiyou 117, and inbred indica rice Hainonghong 1, Meixiangzhan, Guiyu 9, Huaxiang, and Wushansimiao—were selected Provinces Guangdong, Guangxi, and Hanan, China. Japonica rice Nipponbare (NIP) and a local indica rice variety, U25, were used as high-susceptibility and high-resistance controls, respectively. SRBSDV and WBPHs were collected from fields in Luoding, Guangdong Province, China, were maintained on NIP plants in a greenhouse, and used in all experiments in this study.

### 2.2. Virus Inoculation Simultaneously for Multiple Varieties

Seeds of the tested rice varieties were soaked in water for 12 h, germinated for 2 days at 37 °C, and then seeded in a 29 cm diameter circular flowerpot with organic peat soil (Jiffy) as the substrate. After the seedlings reached the 2-leaf stage, the weak seedlings were removed, and 20 healthy rice seedlings were retained in each pot. Then, 2 pots for each variety, 24 pots in total, were randomly arranged in a small net room. At the same time, the WBPH nymphs at the 3rd–4th instar fed on SRBSDV-infected rice plants at the tillering stage for over 12 days, by which time the hoppers had developed to the adult stage; then, an RT-PCR detection for SRBSDV was conducted following Yang et al. [4]. Only those hopper populations with over 80% SRBSDV positivity were used for subsequent experiments. When the tested rice seedlings grew to the 3–4-leaf stage, the viruliferous WBPH adults were released into the net room. The number of hoppers was about 2 individuals for each rice plant. After 5 days of coculture, the virus was inoculated, and the WBPHs were removed, and the rice seedlings were kept in the net room to grow to the 4–5-leaf stage. The conditions for rice and WBPH development were 25 °C–30 °C, 80%–90% relative humidity, and 13 h and 11 h light and dark alternation. Finally, all tested seedlings were individually transplanted in the field until they grew to the elongation stage for symptom observation. The experiments were conducted in Guangzhou, China from June to September 2020, with three replicates, each variety of which was inoculated with no less than 30 plants.

### 2.3. Virus Inoculation Separately for Single Varieties

The seedlings of the tested rice varieties were cultivated by the above methods. Each pot containing 20 seedlings at the 3–4-leaf stage was covered by a small plastic cup with some vents punched in the side. A total of 10 viruliferous WBPH adults were released into each cup. After 2 days of coculture, the insects were removed, and the rice seedlings were cultured to the 4–5-leaf stage and then transplanted in the field for symptom observation. The experiments were repeated three times, and more than 30 seedlings were tested for each variety in each experiment.

### 2.4. Observation of Symptom Severity

At 45 days post-inoculation (dpi), the tested rice plants had developed to the elongation stage and were observed. SRBSDV-infected plants were diagnosed based on the symptoms of dwarfism and dark green color and the results of RT-PCR detection. The plants showing SRBSDV symptoms and RT-PCR positivity were defined as infected plants; others were defined as healthy plants. All tested plants were observed, and the SRBSDV infection rate was calculated for each variety. Meanwhile, over 10 infected, and healthy plants for each variety were measured for their height and fresh weight. The relative dwarfing rate and the relative weight loss rate of the infected plants were calculated and compared with the healthy plants.

### 2.5. Measurement of Virus Accumulation in Infected Plants

At 15 dpi, 30 dpi, and 45 dpi, tissues from the second developed upper leaf of infected and healthy plants were sampled and detected via RT-qPCR [15] and Western blot [16] for the SRBSDV titer. At least five infected and healthy plants were detected for each variety, and the relative virus accumulation level was determined by the 2 delta–delta Ct method and t-test analysis or one-way ANOVA SPSS19.0 software (IBM, New York, NY, USA).

### 2.6. Determination of the WBPH Survival Rates on Different Rice Varieties

To eliminate any influence of hopper resistance on virus infection, the survival rates of WBPH on different varieties were determined. In total, 15 to 20 WBPH adults were raised on 5 plants at the 3–4-leaf stage in a glass tube. Two days later, the hoppers were checked, and their survival rates on different rice varieties were calculated. The conditions of the culture were the same as above. The experiment was repeated three times.

### 2.7. Evaluation of WBPH Virus Acquisition Ability from Infected Rice Plants

Approximately 30 WBPH adults were exposed to SRBSDV-infected plants at 15 dpi, 30 dpi, and 45 dpi in a cage for 6 h. The hoppers were then transferred to healthy rice seedlings to raise for 15 days, then detected individually by RT-PCR for SRBSDV [4]. The virus acquisition rates of the WBPHs from the diseased plants of different varieties were calculated and compared. The experiment was repeated three times.

### 2.8. Date Analysis

The statistical software package SPSS 19.0 (IBM, Armonk, NY, USA) was used to analyze the experimental data. The data were analyzed by one-way analysis of variance (ANOVA) to check for significant differences between infection rates in different varieties, virus accumulation in infected plants, WBPH survival rates on different rice varieties, and WBPH virus acquisition potency from infected rice plants at *p* < 0.05 or *p* < 0.01; the dwarfing rate and weight loss were analyzed by Duncan’s multiple range test at *p* < 0.05. Pearson correlation analysis was used to analyze the relationship between virus accumulation and WBPH virus acquisition rate.

## 3. Results

### 3.1. Resistance Based on Infection Rate

In order to detect dominant variety resistance against SRBSDV, we checked the infection rate via two methods: simultaneous virus inoculation for multiple varieties and separate virus inoculation for single varieties. The results obtained by the two methods were basically the same. No immune varieties were found, but the infection rates of different varieties could be divided into three categories (Figure 1A,B). U25, with an infection rate less than 30%, was defined as a highly resistant variety; Guang8you 165, Guiyu 9, and Hainonghong 1, with infection rates between 31% and 60%, were defined as resistant varieties; the remaining eight varieties, including Nip, with infection rates greater than 61%, were defined as susceptible varieties.

To determine whether the differences in the infection rates of varieties were related to insect resistance [17], we measured the WBPH survival rates on the healthy seedlings of different varieties; the results showed no significant difference in the survival rates on the tested varieties except for U25 (Figure 2), indicating that U25 has a degree of resistance to WBPH. The lower infection rate of variety U25 might therefore be a result of a combination of insect resistance and disease resistance. The infection rates of the other varieties reflect their virus resistance.

### 3.2. Resistance Based on Symptom Severity

After SRBSDV infection, typical symptoms of plant dwarfing, dark green leaves, short and stiff leaves, and crinkled, even twisted leaves, were observed on the infected plants of the 12 tested varieties. No significant differences in the types of symptoms were observed among varieties, but the severity of symptoms varied (Figure 3A).

At 45 days post-infection (dpi) with SRBSDV, the plant height and weight of the infected plants of each variety were significantly reduced. The relative dwarfing rate and the relative weight loss rate of infected plants were calculated compared with the healthy plants. Based on these rates, we can divide all varieties into three categories: varieties U25 and Guiyu 9, with dwarfing rates of 30%–45% and weight loss rates less than 70%, were classified as resistant varieties; varieties Nip, Yexiangyou 9, and Hainonghong 1, with dwarfing rates of 40%–55% and weight loss rate of 70%–80%, were defined as moderately resistant varieties; varieties Guiyou 117, Wushansimiao, Huaxiang, Guyou 248, Meixiangzhan, Baixiangyou 9978, and Guang8you 165, with dwarfing rates over 55% and weight loss rates over 80%, were identified as low-resistance varieties (Figure 3B,C).

### 3.3. Resistance Based on Virus Accumulation in Infected Plants

Virus accumulation often reflects the resistance of varieties to a virus. We detected the virus accumulation in plants infected with SRBSDV by RT-qPCR and Western blot. There were significant differences in virus accumulation among the different rice varieties. According to the average virus accumulation in leaf tissues of infected plants at 15, 30, and 45 dpi, the 12 tested varieties can be divided into 3 categories: the varieties Guang8you 165, Guiyu 9, Wushansimiao, Meixiangzhan, Hainonghong 1, and U25 with low virus accumulation; the varieties Huaxiang, Guyou 248, Guiyou 117, and NIP with medium virus accumulation; the varieties Yexiangyou 9 and Baixiangyou 9978 with high virus accumulation. The difference in virus accumulation between resistant and susceptible varieties was more than sixfold. Additionally, there were significant differences in virus accumulation patterns among the different varieties. The varieties with low virus accumulation also showed slow virus titer increase with the development of the disease course, which indicated that these varieties had strong resistance to SRBSDV propagation. For example, varieties Guang8you 165, Guiyu 9, Wushansimiao, and NIP supported rapid virus accumulation that reached a peak at 30 dpi, then decreased; variety Guiyou 117 had higher virus accumulation in the early stage of virus infection, and the accumulation gradually decreased with the development of the disease course; variety Baixiangyou 9978 accumulated the virus to high levels at an early stage following a sharp decrease and increase; variety Yexiangyou 9 had high virus titers only in a short middle period of the course of the disease (Figure 4A,B). The above results were verified by Western blotting (Figure 4C).

### 3.4. Resistance Based on Insect Vector Virus Acquisition

Virus acquisition by the insect vector (WBPH) from infected rice plants determines how quickly SRBSDV can spread and become epidemic [18,19]. Therefore, we compared the WBPH virus acquisition potencies from infected plants of different rice varieties. Generally, the virus acquisition rates of WBPH from the infected plants were low at the early infection stage and increased with the disease course (Figure 5A). There was a significant positive correlation between WBPH virus acquisition rates and the virus accumulation levels in infected rice plants (r = 0.624, *p* = 0.03 < 0.05). Based on the mean virus acquisition rates of WBPHs at the three disease time points (15, 30, and 45 dpi), we can group the tested varieties into three categories: varieties Guang8you 165, Guiyu 9, and U25, had low virus supply ability; varieties Wushansimiao, Meixiangzhan, and Hainonghong 1 had moderate virus supply ability; varieties Huaxiang, Guiyou 117, Guyou 248, NIP, Yexiangyou 9, and Baixiangyou 9978 had high virus supply ability (Figure 5B).

## 4. Discussion

In previous reports, infection rate and severity of symptoms were usually used to evaluate the disease resistance of crop varieties. However, crop resistance should include characteristics that slow down a disease epidemic. The concept of slow rusting for wheat rust resistance was put forward by Caldwell [20] and applied successfully to breeding wheat-rust-resistant varieties [21,22,23].

In this study, we evaluated the SRBSDV resistance of 10 dominant rice varieties from southern China, where the virus overwinters and accumulates as a source of early spring reinfection, by artificial inoculation. We checked the varieties for four characteristics: infection rate, symptom severity, virus accumulation in infected plants, and WBPH virus acquisition potency from infected rice plants. Among these four characteristics, only the infection rate is directly related to the yield loss of rice caused by the disease and reflects the disease resistance of varieties in the traditional sense. The virus accumulation in infected plants and WBPH virus acquisition potency from infected plants represent a typical case of slow diseasing resistance. While the symptom severity regarding dwarfing and weight loss is not associated with rice yield loss due to the infected plants being unable to ear and produce grain, the degree of dwarfing and the reduction in the fresh weight of diseased plants have a deleterious effect on the living space and food quantity of the virus-transmitting vector WBPH; therefore, symptom severity is also a characteristic of slow disease resistance.

Among the 10 dominant varieties tested in this study, variety Guiyu 9 showed resistance in the above 4 characteristics; the two varieties Guang8you 165 and Hainonghong 1 showed resistance in the infection rate, virus accumulation, and WBPH virus acquisition ability; and the two varieties Meixiangzhan and Wushansimiao showed resistance in virus accumulation and WBPH virus acquisition ability. These varieties can be recommended for use in production to reduce losses due to disease and slow the spread of disease.

In Asia, WBPH, the vector of SRBSDV, is a typical long-distance migratory insect. In spring, it migrates from low latitude to high latitude with the southeast monsoon [24], resulting in the rapid spread of the virus on a large scale; then, in autumn, it migrates from high latitude to low latitude with the northeast monsoon, carrying the virus into Vietnam and southern China [6,7]. In addition, WBPH can proliferate and can acquire and transmit SRBSDV in a short time, resulting in frequent reinfection of SRBSDV [25]. Although the infection period of severe yield loss caused by SRBSDV is the rice seedling stage [8], the rational utilization of disease-resistant varieties, including those with slow disease resistance, in southern China, where the virus overwinters and accumulates as a source in early spring, can reduce the yield loss caused by the disease in local rice and slow the disease epidemic in high-latitude areas.

## 5. Conclusions

This study confirmed that the resistance of different rice varieties to SRBSDV exhibits diverse properties, including reducing virus infection and slowing disease epidemics. Although there was no immune resistance found in southern China’s dominant rice varieties, some indications of infection resistance and disease-slowing resistance were screened. Rice varieties with low virus accumulation levels and vector virus acquisition ability in infected plants can be used to minimize the source of reinfection and slow the spread of the disease.

## Figures and Tables

**Figure 1 viruses-13-01501-f001:**
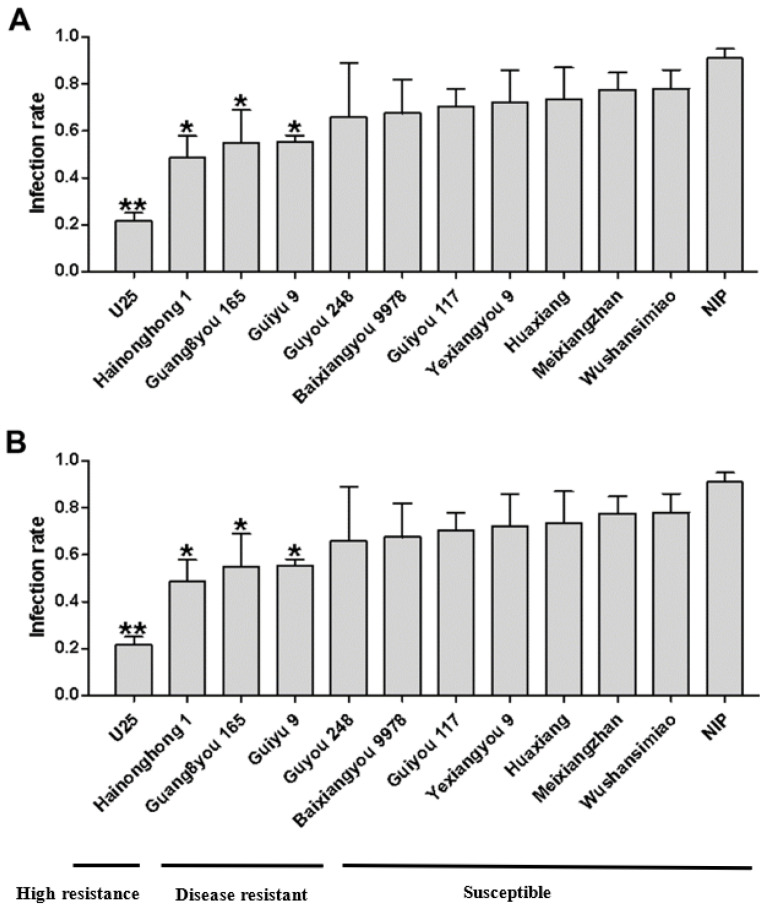
Infection rates of dominant rice varieties in southern China at the seedling stage. The results are shown as follows: (**A**) simultaneous virus inoculation for multiple varieties; (**B**) separate virus inoculation for single varieties; each variety was compared to NIP, “*” indicates a significant difference (*p* < 0.05), “**” indicates extremely significant difference (*p* < 0.01).

**Figure 2 viruses-13-01501-f002:**
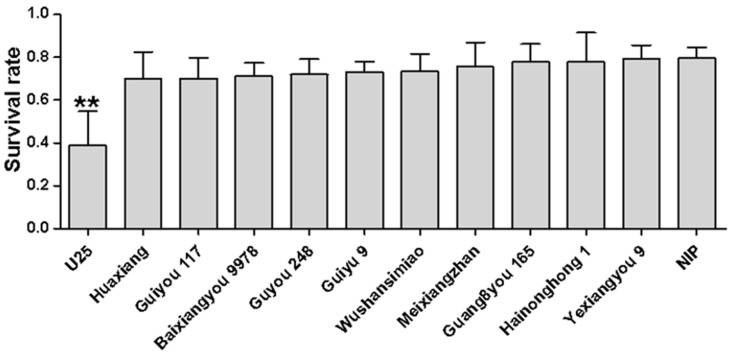
Determination of WBPHs survival rates on the seedlings of different rice varieties. Each variety was compared to NIP, “**” indicates an extremely significant difference (*p* < 0.01).

**Figure 3 viruses-13-01501-f003:**
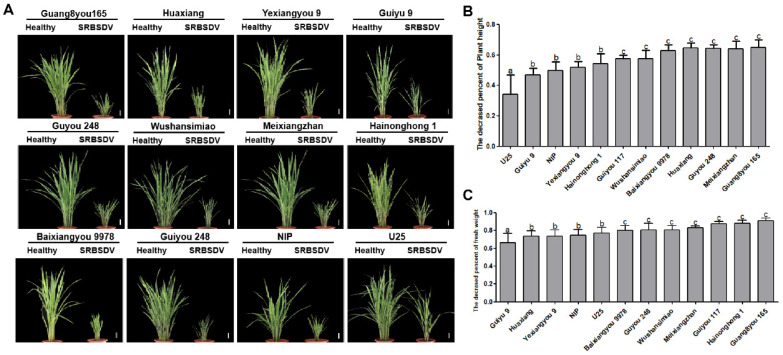
Symptoms (**A**), dwarfing rates (**B**), and weight loss rates (**C**) of plants at 45 days post-infection with SRBSDV. Lowercase letters indicate significant differences (*p* < 0.05).

**Figure 4 viruses-13-01501-f004:**
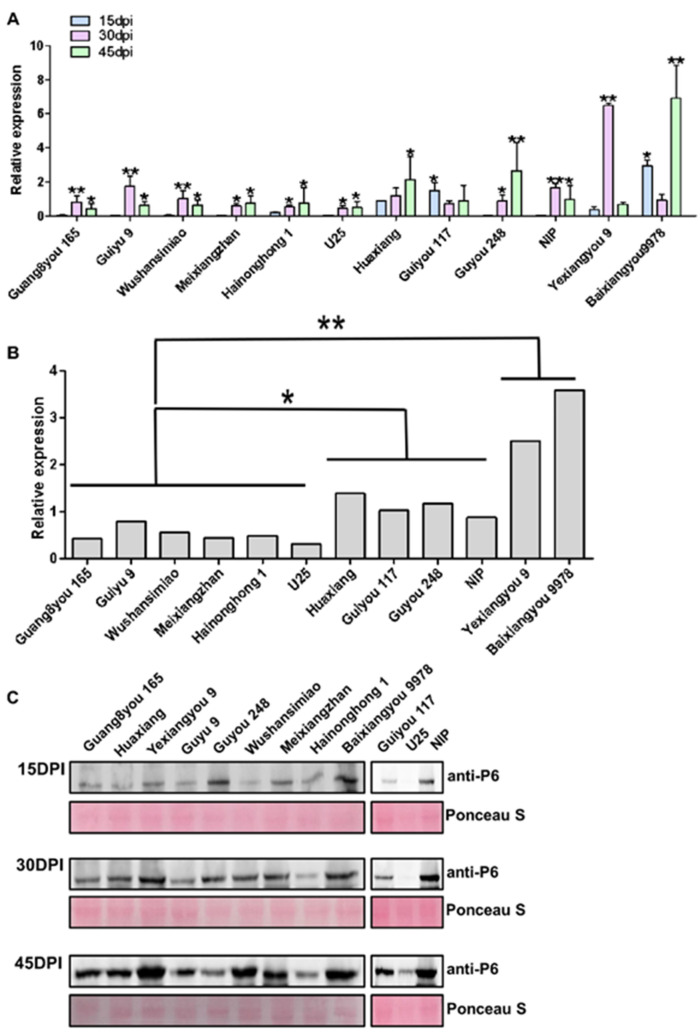
The virus accumulation in the plants infected with SRBSDV at 15, 30, and 45 dpi: (**A**) RT-qPCR analysis for the transcription level of SRBSDV-S10; (**B**) the mean virus titers of three disease stages in infected plants; (**C**) Western blot detection for the accumulation level of SRBSDV-P6. “*” indicates a significant difference (*p* < 0.05), “**” indicates an extremely significant difference (*p* < 0.01).

**Figure 5 viruses-13-01501-f005:**
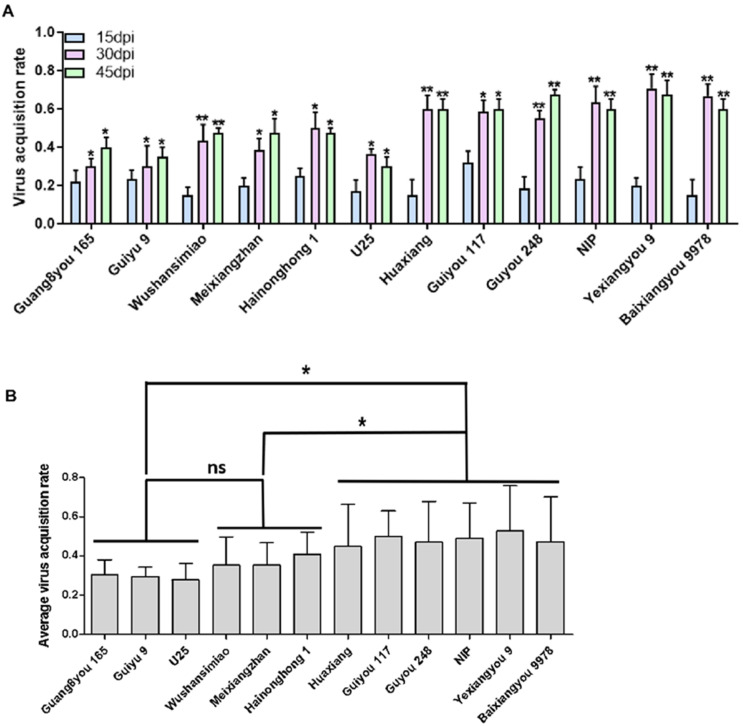
WBPH virus acquisition rates from infected rice plants: (**A**) the rates at which WBPHs acquired SRBSDV from infected rice plants; (**B**) the mean rates of WBPH virus acquisition from infected rice plants at three disease stages. “ns” means no significant difference, “*” means a significant difference (*p* < 0.05), “**” means an extremely significant difference (*p* < 0.01).

## Data Availability

Not applicable.

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
