# Peer review of "Resistance Evaluation of Dominant Varieties against Southern Rice Black-Streaked Dwarf Virus in Southern China"

_viruses, 2021, doi:10.3390/v13081501_

Round 1

Reviewer 1 Report

The manuscript of Zhou et al reports a highly interesting study on the evaluation of rice varieties for SRBSDV resistance. This study is performed at a relatively limited scale (which can be explained by the insect mediated transmission of SRBSDV) but is one of the first on this subject of main interest for rice culture.

However, some points severely impact the quality of the manuscript and have to be corrected or taken in consideration in the writing or interpretation of results before publication.

The  quality of english language and style is poor and some vocabulary terms do not correspond to the standard of the field. I suspect the editorial quality to explain the apparent contradiction in paragraph 138-145 between the sentences « the susceptible rates of different varieties were basically the same » (lines 138-139) and « the 12 varieties could be divided into three categories » (line 141). As other examples (not exhaustive) I noticed unclear sentences or sentences that have to be corrected lines 17-21, 43-45, 54-58, 77-78.

Susceptible rate” should be replaced by “infection rate“ or “incidence”. « Tolerance » generally refers to the capacity of a plant to support the same amount of virus (or pathogen in general) with lower yield loss (or more generally lower impact on plant development). Based on this definition, this term is not adapted to describe varieties that do not accumulate a high amount of virus (paragraph 3.3) and should be used more cautiously to describe the phenotype of plants in part 3.2 (as comparison of Fig 3 and Fig 4 suggested that a lower impact on plant development is associated to a lower virus content, at least for some varieties).

The source of virus and of WBPH is not clearly presented. Lines 67-69 suggested that infected plants with WBPHs collected in Luoding are used. If so, is the same source of inoculum (from the same plants) used throughout all the experiments ? How are the plants and the insects maintened in the greenhouse ? If the inoculum sources result from different field collects, how could you be sure that the viral isolate/the viral content/the WBPHs strain are the same in the different experiments ? For the evaluation of WBPHs virus acquisition, where are the insects coming from ? How could you be sure they have not been infected before the experiment ? Although, depending of the pathosystem, it may be difficult to proceed more rigourously, this point should be discussed. Additional methodological points have to be more clearly described :

- line 82 : are all the plants transplanted or just the ones that are positive based on RT-PCR ? Considering that symptom severity and virus accumulation are measured on transplanted plants (which should be more clearly indicated), are the non-infected plants (if they are transplanted) included in symptoms and virus accumulation ?

- line 93 : what does « pest control cultivation » mean : phytosanitary treatments against fungi and insects ? SRBSDV monitoring ?

- how many plants are transplanted after inoculation of the seedlings treated as described in 2.3 ?

- were symptom observation and virus accumulation monitored on plants inoculated as described in 2.2, in 2.3, or indifferently on both ?

- control plants have been used to estimate the dwarfing rate and the weigth loss, but how and where have the control plants been cultivated ? Have they been tested for absence of virus ? If so how ? If not, could you be sure that there are virus free ?

- virus accumulation: was a single plant tested for each replicate ? Or a pool of plants ? What is exactly the « second expanding leaf » ?

The satistical analysis of the results is not correctly described and could probably be improved. For dwarfing rate and weight loss, significantly different groups of varieties are mentionned (Fig3B&C), which suggested that ANOVA and post-hoc tests have been performed, but the post-hoc test that has been used should be mention in 2.8 and in the legend of Fig 3. For most of the other bar charts, the significant differences indicated by « * » suggested that pairwise comparisons have been performed. Although it can be supposed that each variety was compared to NIP, this should be more clearly indicated. In addition, ANOVA and post-hoc tests seem to be more adapted to analyse this sort of data set, why did you prefer pairwise comparisons ?

As additional minor comments, I would suggest to add a figure on the correlation between the amount of virus and the virus acquisition by WBPH. In the discussion (line 242), I did not understand why « the susceptible (infection) rate...reflects the disease resistance of varieties in the traditional sense ». Indeed, while « resistance » may be used to define very different cases, I would say it most often refers to the effect of major R genes that control a complete absence of pathogen developement (except in case of breakdown), which is not the case here.

Author Response

C1. The manuscript of Zhou et al reports a highly interesting study on the evaluation of rice varieties for SRBSDV resistance. This study is performed at a relatively limited scale (which can be explained by the insect mediated transmission of SRBSDV) but is one of the first on this subject of main interest for rice culture.

R1: Thank you for your kind comments. It is true that our study was a relatively limited scale but included major varieties in southern China rice-growing areas. The fact is that the study of insect-transmitted virus is a lot of work and time consuming.

C2. However, some points severely impact the quality of the manuscript and have to be corrected or taken in consideration in the writing or interpretation of results before publication.

- The quality of English language and style is poor and some vocabulary terms do not correspond to the standard of the field. I suspect the editorial quality to explain the apparent contradiction in paragraph 138-145 between the sentences «the susceptible rates of different varieties were basically the same» (lines 138-139) and «the 12 varieties could be divided into three categories» (line 141). As other examples (not exhaustive) I noticed unclear sentences or sentences that have to be corrected lines 17-21, 43-45, 54-58, 77-78.

R2: The manuscript was thoroughly revised, and edited by a native English speaker. We hope it is clear enough to meet your comments.

C3. “Susceptible rate” should be replaced by “infection rate” or “incidence”. «Tolerance» generally refers to the capacity of a plant to support the same amount of virus (or pathogen in general) with lower yield loss (or more generally lower impact on plant development). Based on this definition, this term is not adapted to describe varieties that do not accumulate a high amount of virus (paragraph 3.3) and should be used more cautiously to describe the phenotype of plants in part 3.2 (as comparison of Fig 3 and Fig 4 suggested that a lower impact on plant development is associated to a lower virus content, at least for some varieties).

R3: We have replaced “Susceptible rate” with “infection rate”. We agree your suggestion about the concept “tolerance”, and redefined resistance in the revision.

C4: The source of virus and of WBPH is not clearly presented.

- Lines 67-69 suggested that infected plants with WBPHs collected in Luoding are used. If so, is the same source of inoculum (from the same plants) used throughout all the experiments?

R4: Thank you for pointing out the deficiency. The statement now reads as “SRBSDV and WBPHs were collected from fields in Luoding, Guangdong Province, China, maintained on NIP plants in a greenhouse, and used in all experiments in this study” in line 82-85.

- How are the plants and the insects maintained in the greenhouse?

R: We have indicated that “The conditions for rice and WBPH development were 25℃-30℃, 80%-90% relative humidity, and 13 h and 11 h light and dark alternation” in line 104-107.

- If the inoculum sources result from different field collects, how could you be sure that the viral isolate/the viral content/the WBPHs strain are the same in the different experiments?

R: The inoculum sources were collected from a same field. We mentioned this in line 82-85.

- For the evaluation of WBPHs virus acquisition, where are the insects coming from?

R: WBPHs were collected from the field in Luoding, Guangdong Province, China. We mentioned this in line 82-85.

How could you be sure they have not been infected before the experiment?

R: The WBPH adults were collected from a healthy field in Luoding and transferred to healthy rice plants in greenhouse for propagation. To ensure the insects were SRBSDV free, some of them were detected by RT-PCR before the experiment.

C5: Line 82: are all the plants transplanted or just the ones that are positive based on RT-PCR?

The tested seedlings at least 30 for each variety are transplanted whether or not infected. It reads as “all tested seedlings were individually transplanted in the field until they grew to the elongation stage for symptom observation. The experiments were conducted in Guang-zhou, China from June to September 2020, with three replicates, each variety of which was inoculated with no less than 30 plants” in line 107-114 in the revision.

Considering that symptom severity and virus accumulation are measured on transplanted plants (which should be more clearly indicated), are the non-infected plants (if they are transplanted) included in symptoms and virus accumulation?

R: As descripted in line 133-136 in the revision, “The plants showing SRBSDV symptoms and RT-PCR positivity were defined as infected plants; others were defined as healthy plants. All tested plants were observed, and the SRBSDV infection rate was calculated for each variety. Meanwhile, over 10 infected and healthy plants for each variety were measured for their height and fresh weight”.

C6: line 93: what does «pest control cultivation» mean: phytosanitary treatments against fungi and insects? SRBSDV monitoring?

R: We deleted this statement in the revision. The tested rice was managed in conventional ways.

C7: How many plants are transplanted after inoculation of the seedlings treated as described in 2.3?

R: As descripted in line 107-114 in the revision. All tested seedlings were transplanted.

C8: were symptom observation and virus accumulation monitored on plants inoculated as described in 2.2, in 2.3, or indifferently on both?

R: Yes. This was indicated in line 107-108 in the revision as “all tested seedlings were individually transplanted in the field until they grew to the elongation stage for symptom observation”.

C9: control plants have been used to estimate the dwarfing rate and the weight loss, but how and where have the control plants been cultivated?

R: As descripted in the revision, “The plants showing SRBSDV symptoms and RT-PCR positivity were defined as infected plants; others were defined as healthy plants”(line 133-134), we use the healthy plants as control, then “The relative dwarfing rate and the relative weight loss rate of the infected plants were calculated compared with the healthy plants”(line 137-138).

Have they been tested for absence of virus? If so how? If not, could you be sure that they are virus free?

R: It reads now in the revision as “At 45 days post-inoculation (dpi), the tested rice plants had developed to the elongation stage and were observed. SRBSDV-infected plants were diagnosed based on the symptoms of dwarfism and dark green color and the results of RT-PCR detection. The plants showing SRBSDV symptoms and RT-PCR positivity were defined as infected plants; others were defined as healthy plants” (line 130-134).

C10: virus accumulation: was a single plant tested for each replicate? Or a pool of plants? What is exactly the «second expanding leaf»?

R: As descripted in revision, “At 15dpi, 30dpi and 45dpi, tissues from the second developed upper leaf of infected and healthy plants were sampled and detected via RT-qPCR [15] and Western blot [16] for the SRBSDV titer. At least five infected and healthy plants were detected for each variety, and the relative virus accumulation level was determined” (line 145-149).

C11: The statistical analysis of the results is not correctly described and could probably be improved. For dwarfing rate and weight loss, significantly different groups of varieties are mentioned (Fig3B&C), which suggested that ANOVA and post-hoc tests have been performed, but the post-hoc test that has been used should be mention in 2.8 and in the legend of Fig 3. For most of the other bar charts, the significant differences indicated by « * » suggested that pairwise comparisons have been performed. Although it can be supposed that each variety was compared to NIP, this should be more clearly indicated.

R: We rewrote the section 2.8 (line 180-190 in revision). Hope it is clear and appropriate.

In addition, ANOVA and post-hoc tests seem to be more adapted to analyse this sort of data set, why did you prefer pairwise comparisons?

R: Thank you for your suggestion. We believe the results can be showed clearly when all tested varieties being compared with one control variety NIP. Hope it can be agreed by you.

C12: As additional minor comments, I would suggest to add a figure on the correlation between the amount of virus and the virus acquisition by WBPH. In the discussion (line 242), I did not understand why «the susceptible (infection) rate...reflects the disease resistance of varieties in the traditional sense». Indeed, while «resistance» may be used to define very different cases, I would say it most often refers to the effect of major R genes that control a complete absence of pathogen development (except in case of breakdown), which is not the case here.

R12: Thank you for the suggestions. The data of the virus accumulation and the virus acquisition by WBPH were showed in figure 4 and 5. To avoid repetition, we did not add another figure. We gave a statement as “There was a significant positive correlation between WBPH virus acquisition rates and the virus accumulation levels in infected rice plants (r=0.624, p=0.03<0.05)” in the revision (line 307-309).

As you pointed out, “resistance” may have different meanings in different case, in our case, the virus accumulation and the virus acquisition by WBPH might be included in the concept of resistance. Therefore, we rewrote the discussion section in the revision. We suggested “the virus accumulation in infected plants and WBPH virus acquisition potency from infected plants represent a typical case of slow diseasing resistance” (line 347-349). We hope you agree with us.

Reviewer 2 Report

In a general way, the manuscript is poorly written. Very important informations are missing. Just to exemplify, even the virus taxonomy or virus genome is cited. In the way the manuscript is presented is almost impossible to understand how the data were obtained. 

Author Response

The manuscript was thoroughly revised, and edited by a native English speaker. Hope it was improved enough to read clearly. We thank you very much for you comments and will write our manuscript more carefully in future.

Reviewer 3 Report

In this paper the authors evaluate the resistance to Southern rice black-streaked dwarf virus (SRBSDV), a Fijivirus of Reoviridae family. SRBSDV was first reported in China as a serious pathogen of rice. The resistance to SRBSDV of rice varieties showed no immune resistance but some differences in the susceptibility and symptoms, not only reducing the damage, but also the re-infection source and the damage to rice in other areas. There was lower virus accumulation in the early stage of disease differing among varieties, with positive correlation between virus accumulation and insect vector acquisition. The autheirs believe that the results of this study can help with the selection of rice varieties for the winter/spring propagation.

This is an usefull paper that could be published, provided other reviewers agreee. The paper requires some modifications, as follows.

1, Abstract needs to be shortened and edited so the information and English both flow smoothly and interestingly.

2,  Introduction OK, first  but the first word in the virus name should begin with capital Southern.

3. Methods and Results are described with adequate detail. 

4. Discussion and Conclusions.  At the end please explain why do you believe your results will help to inhibit the frequent recurrence of the virus.

5.  The entire text needs to be read by a native English speaker and carefully edited, as there are numerous mistakes and grammatical discrepancies.

Author Response

C1: In this paper the authors evaluate the resistance to Southern rice black-streaked dwarf virus (SRBSDV), a Fijivirus of Reoviridae family. SRBSDV was first reported in China as a serious pathogen of rice. The resistance to SRBSDV of rice varieties showed no immune resistance but some differences in the susceptibility and symptoms, not only reducing the damage, but also the re-infection source and the damage to rice in other areas. There was lower virus accumulation in the early stage of disease differing among varieties, with positive correlation between virus accumulation and insect vector acquisition. The autheirs believe that the results of this study can help with the selection of rice varieties for the winter/spring propagation. This is an usefull paper that could be published, provided other reviewers agree.

R1: Thank you for positive comments.

C2: Abstract needs to be shortened and edited so the information and English both flow smoothly and interestingly.

R2: The abstract as well as the whole manuscript was thoroughly revised and edited by a native English speaker.

C3: Introduction OK, first but the first word in the virus name should begin with capital Southern.

R3: Your suggestion has been accepted.

C4: Methods and Results are described with adequate detail.

R4: We rewrote this section and more details were provided in the revision.

C5: Discussion and Conclusions. At the end please explain why do you believe your results will help to inhibit the frequent recurrence of the virus.

R5: These sections were rewritten thoroughly. Hope it response your concern.